# Multiple Criteria Decision-Making in Heterogeneous Groups of Management Experts

**Virgilio López-Morales**

Information Technology and Systems Research Center (CITIS), Universidad Autónoma del Estado de Hidalgo, 42184 Pachuca, Hidalgo, Mexico; virgilio@uaeh.edu.mx; Tel.: +52-771-71-72000 (ext. 6734)

**Abstract:** In commercial organizations operations, frequently some dynamic events occur which involve operational, managerial, and valuable information aspects. Then, in order to make a sound decision, the business professional could be supported by a Multi Criteria Decision-Making (MCDM) system for taking an external course of action, as, for instance, forecasting a new market or product, up to an inner decision concerning for instance, the volume of manufacture. Thus, managers need, in a collective manner, to analyze the actual problems, to evaluate various options according to diverse criteria, and finally choose the best solution from a set of various alternatives. Throughout these processes, uncertainty and hesitancy easily arise, when it comes to define and judge criteria or alternatives. Several approaches have been introduced to allow Decision Makers (DMs) to deal with. The Interval Multiplicative Preference Relations (IMPRs) approach is a useful technique and the basis of our proposed methodology to provide reliable consistent and in consensus IMPRs. In this manner, DMs' choices are implicitly including their uncertainty while maintaining both an acceptable individual consistency, as well as group consensus levels. The present method is based on some recent results and an optimization algorithm to derive reliable consistent and in consensus IMPRs. In order to illustrate our results and compare them with other methodologies, a few examples are addressed and solved.

**Keywords:** uncertain group decision-making support systems; multiple criteria decision-making; reliable group decision-making; interval multiplicative preference relations

## 1. Introduction

Among the vast world of commercial operations organizations and despite their differences, they have common business operations or activities such as acquiring inventory, hiring employees and cashing from customers. Nowadays, inside each modern organization, we can frequently find an information system working in synchrony with these business operations. Furthermore, several important information systems are fed by operating departments (work centers of the organization), and, as a result, these systems' outcomes can be used to manage these operations. As a consequence, managers analyze their corresponding information system in light of the work that the organization performs. For instance, when a marketing manager is required to advise management and to have some reports for management decision-making, she/he must understand the organization's product cycles.

There are various events which occur while organizations engage their business operations as, for instance, diverse trends in purchases and sales. This dynamic data coming from these events are frequently recorded and kept up in a database to mirror and supervise business operations. These records include operational, managerial, and valuable information details. Thus, in order to design and use a group decision-making system, the business professional must previously consider what kind of event data is needed and the necessary process to extract the useful information.

In essence, the type of decision under consideration rules what kind, amount and quality of the information must be used to make a sound decision. Furthermore, information is more valuable when it recognizes and doesn't disregard the personal management styles, the main choices of the the decision maker, the weighting of each decision maker with respect to the entire decision-making process, and managers' uncertainty and hesitancy when a heterogenous group tries to achieve an integral decision.

Multi criteria decision-making then could support human-centered management in taking a course of action for a sound decision-making to elucidate several management problems. For instance, what products to sell and the suitable targeted market to sell them, or the structure of the organization better suited for this process or even the direction and motivation of employees based on some known standards. Another type of management problem can be an inner decision as for instance, human resources information, volume of manufacture and the available delivering chains, which is useful to develop alternative methods for manufacturing and delivering a new product.

On the other hand, the essence of the information provided to managers must be according to the management level. For example, strategic level managers need information allowing them to assess the environment to forecast future events and conditions, where s/he may not be as concerned with the timeliness or accuracy of the information as her/his interest is in the trends. In this case, most of the information is exogenous to the organization, or tactical management needs information mainly coming from pertinent operational units. Some exogenous information is needed, as well as more detailed and accurate information than the information required at the strategic level. Finally, operational management unit usually requires exogenous narrower information but more detailed and accurate information. It comes largely from within the organization where frequent decisions are made, with shorter lead times to respond in a timely manner to current variations as, for instance, in sales patterns.

As we noted in this section, managers support the entire firms' making decision to achieve diverse kind of goals. They have basically different kinds of perspectives and local goals to reach a high efficiency level at their respective departments. However, the main goal of the firm needs, at certain moments, to combine local goals and perspectives of the work centers, with particular global targets.

In this scenario of heterogeneous managers group, an MCDM system for supporting the management for group decision-making becomes paramount.

Fortunately, there are a vast variety of MCDM techniques ranging from: analytic hierarchy process (AHP) [1], multi attribute utility theory (MAUT) [2], simple multi attribute rating technique (SMART) [3], fuzzy set theory (FST) [4], data envelopment analysis (DEA) [5], case-based reasoning (CBR) [6], simple additive weighting (SAW) [7], elimination et choice translating reality (ELECTRE) [8], technique for order of preference by similarity to ideal solution (TOPSIS) [9], preference ranking and organization method for enrichment evaluation (PROMETHEE) [10], and goal programming (GP) [11]. Another interesting method for addressing uncertainty is the Hesitant Fuzzy Sets (HFS) [12] where some recent contributions for heterogeneous information are found [13–18]. These methods are appropriate for uncertainty problems, since fuzzy logic aims to represent human preferences based on individual opinions expressed through a linguistic setting. Its major drawback is that membership functions (which can be seen as intervals) are fixed and also, until now, there does not exist detailed work related to an appropriate analytical tool (norms, aggregation operators, etc.) for a particular study case. Thus, it results in a greater uncertainty and vagueness to the problem.

Recently, several frameworks have been employed and successfully applied to solve decision problems in many areas, including international politics and laws [19], transportation [20–23], business intelligence [24], information and communication technologies [25], water resources management [26], environmental risk analysis [27], flood risk management [28], environmental impact assessment and environmental sciences [14,29], solid waste management [30], climate change [31], remote sensing [32], energy [33], health technology assessment [34] and nanotechnology research [35]. Furthermore, MCDM techniques have been integrated with known systems such as genetic algorithms,

geographic information systems, fuzzy logic and intelligent systems, automatic control systems and neural networks which recently are being applied.

Group Decision Making (GDM) is a main MCDM issue, where multiple DMs (managers in our case) act collaboratively and collectively, analyze decision-making problems, evaluate goals according to a set of criteria, and finally choose the best solution from a set of alternatives [36]. As noted in [37], when organizations gather specialized groups or larger groups, and the number of alternatives increase, unanimity may be difficult to attain. For this reason, flexible or milder benchmarks (definitions) of Group Consensus and Individual Consistency have been employed.

Group Consensus (GC) has to do with group cooperation and agreement since the alternative, option, or goal to be achieved is the best course of action for the whole organization. On the other hand, Individual Consistency (IC) concerns each DM to have her/his information, and, consequently, her/his judgments, free of contradictions.

Derived of the blend of heterogenous DM group, a common problem in big organizations is that managers often can accurately state definitions and assessments on the priority rate of the set of criteria or alternatives, when it concerns their own operating departments. However, when it comes to define and judge any other set of criteria or alternatives, they face uncertainty and hesitation problems.

In this paper, we address and solve these problems by allowing the DM to utilize blended assessments. For example, DMs can use crisp values for her/his ratio judgments of criteria or alternatives when s/he is confident; and then they can use intervals where these are used to express her/his uncertainty and hesitant assessments.

The aim of this paper is to synthesize a novel approach in order to provide a reliable measure of both the IC and the GC of a set of these blended assessments, which we called Interval-Multiplicative Preference Relations (I-MPRs). Then, in the next step, the approach is verified by a constrained optimization algorithm where the improved I-MPRs will finally fulfill the Individual Consistency and Group Consensus Indices since both restrictions are involved in the constraints.

In order to verify the requirements of acceptable IC and GC levels, we use the Hadamard's operator. As soon as IC and then GC are validated, an Interval Priority Vector can be obtained from this set of ordered judgment I-MPRs, in order to rank the alternatives as a final result of the DM analysis. Various techniques are revised and the prioritization method which indicates the entire order of the intervals and the preference degrees, is addressed according to our results.

The main advantages of our approach follow:

- It is provided through a couple of algorithms and a nonlinear optimization approach concurrently applied.
- Through the Hadamard's operator and some easy algebraic manipulations, objective functionals are synthesized (as it will be detailed further on), to be used in the optimization algorithm.
- When the I-MPRs improved by the methodology are reduced into an MPR (defined in the I-MPR), our approach can still give reliable results. For example, for this MPR, we can verify the results of IC or GC, with an alternative method.
- The IC or the GC accepted indices (threshold values) have been previously investigated and fixed. Nevertheless, the project designer could assign a different value depending on the project requirements.
- Obtained results are independent of the method of prioritization utilized in the consensus operation.

This paper is organized as follows: In Section 2, some preliminaries (Definitions, Theorems and Lemmata) are given to support a basis of the main methodologies and techniques previously described above and the approach introduced here. In Section 3, an extension of analysis and results derived in the former section is used in the I-MPR framework for obtaining our main results. Then, in Section 4, a slight modification of a prioritization method is given in light of the results. In Section 5, some numerical examples are solved and compared with other methodologies. Finally,

in Section 6, concluding remarks are given about the main advantages, drawbacks and future research of our methodology.

## 2. Preliminaries

Let us consider a Group Decision-Making problem and let $D = \{d_1, d_2, \cdots, d_m\}$ be the set of DMs, and $C = \{c_1, c_2, \cdots, c_n\}$ be a finite set of criteria (or alternatives), where $c_i$ denotes the $i$th criteria.

In the AHP framework [1], a pairwise comparison matrix or an MPR is given by a DM where s/he provides judgments through a ratio $(c_i/c_j)$ for every pair of criteria (or alternatives) ($c_i$ and $c_j$) to represent the preference degree of the first criteria (or alternative) over the second. A Saaty's Scale is frequently used to pick up a value for this preference ratio where $SS = [1/9 \ 9]$.

Thus, an MPR for instance A $= (a_{ij})_{n\times n}$, is a positive reciprocal $n \times n$ matrix, $a_{ij} > 0$, such that $a_{ji} = 1/a_{ij}$, $a_{ii} = 1$, $\forall i, j \in N$. Note that $a_{ij} \in [1/9 \ 9]$.

Let $\lambda = \{\lambda_1, \cdots, \lambda_m\}$ be the weight vector of the $m-th$ DM, where $\lambda_s > 0$, $s \in M$, $\sum_{s=1}^{m} \lambda_s = 1$, which can be derived with several techniques (see for instance [38] and the references cited therein).

An MPR $n \times n$ matrix is called a *completely consistent* MPR (cf. [1]) if

$$a_{ij} = a_{il}a_{lj}, \quad \forall i, j, l \in N. \tag{1}$$

Thus, their corresponding completely consistent MPR $K = (k_{ij})_{n\times n}$ can be constructed from each MPR $n \times n$ matrix as follows:

$$k_{ij} = \prod_{l=1}^{n} \left(a_{il}a_{lj}\right)^{1/n} = a_{ij}^{2/n} \prod_{\substack{l=1 \\ i\neq l, \ j\neq l}}^{n} (a_{il}a_{lj})^{1/n}, \tag{2}$$

where $i = 1, 2, \cdots, n-1, j = i+1, \cdots, n$.

Furthermore, let $A^c$ be the group MPR (cf. [39]) which represents the group opinion utilizing the geometric average operator:

$$A^c = (a_{ij}^c)_{n\times n} = \prod_{t=1}^{m} \left(a_{ij}^{(t)}\right)^{\lambda_t}, \quad i, j \in N; \quad t \in M. \tag{3}$$

Let us denote (cf. [40]), an I-MPR $A_t$ given by the $t-th$ expert as

$$A_t = (a_{ij}^{(t)})_{n\times n} = \begin{pmatrix} 1 & \left[\overset{-(t)}{a}_{12} \ \overset{+(t)}{a}_{12}\right] & \cdots & \left[\overset{-(t)}{a}_{1n} \ \overset{+(t)}{a}_{1n}\right] \\ \left[\overset{-(t)}{a}_{21} \ \overset{+(t)}{a}_{21}\right] & 1 & \cdots & \left[\overset{-(t)}{a}_{2n} \ \overset{+(t)}{a}_{2n}\right] \\ \vdots & \cdots & \ddots & \vdots \\ \left[\overset{-(t)}{a}_{n1} \ \overset{+(t)}{a}_{n1}\right] & \left[\overset{-(t)}{a}_{n2} \ \overset{+(t)}{a}_{n2}\right] & \cdots & 1 \end{pmatrix}, \tag{4}$$

where $\overset{-(t)}{a}_{ij}, \overset{+(t)}{a}_{ij} > 0$, $\overset{-(t)}{a}_{ij} \leq \overset{+(t)}{a}_{ij}$, $\overset{-(t)}{a}_{ij} = 1/\overset{+(t)}{a}_{ji}$ and $\overset{+(t)}{a}_{ij} = 1/\overset{-(t)}{a}_{ji}$. Furthermore, $A_t = [\bar{A}_t \ \overset{+}{A}_t]$. For example,

$$\bar{A}_t = (\overset{-(t)}{a}_{ij})_{n\times n} = \begin{cases} \overset{-(t)}{a}_{ij}, & i < j, \\ 1, & i = j, \\ \overset{+(t)}{a}_{ij}, & i > j, \end{cases} \quad \overset{+}{A}_t = (\overset{+(t)}{a}_{ij})_{n\times n} = \begin{cases} \overset{+(t)}{a}_{ij}, & i < j, \\ 1, & i = j, \\ \overset{-(t)}{a}_{ij}, & i > j. \end{cases} \tag{5}$$

## 2.1. Measuring the Dissimilarity between Matrices

A useful operator to measure the degree of dissimilarity between two MPRs is the Hadamard Product (HP). The HP of $A = (a_{ij})_{n \times n}$ and $B = (b_{ij})_{n \times n}$ is defined by

$$C = (c_{ij})_{n \times n} = A \circ B = a_{ij} b_{ij}. \tag{6}$$

Consequently, the degree of dissimilarity between $A$ and $B$ is defined as $d(A, B) = \frac{1}{n^2} e^T A \circ B^T e$ or:

$$\frac{1}{n^2} \Sigma_{i=1}^n \Sigma_{j=1}^n a_{ij} b_{ji} = \frac{1}{n} \left[ \frac{1}{n} \Sigma_{i=1}^{n-1} \Sigma_{j=i+1}^n \left( a_{ij} b_{ji} + a_{ji} b_{ij} \right) + 1 \right], \tag{7}$$

where $e = (1, 1, \cdots, 1)_{n \times 1}^T$. Note that $d(A, B) \geq 1$, $d(A, B) = d(B, A)$ and $d(A, B) = 1$ if and only if $A = B$.

In order to have an assessment of Individual Consistency ($CI$), one can measure the compatibility of $A_l$ with respect to (w.r.t.) its own completely consistent matrix $K$ given by Equation (2). Thus,

$$CI_K(A_l) = d(A_l, K) \leq \overline{CI}, \tag{8}$$

where $\overline{CI} = 1.1$, $A_l$, $l = 1, 2, \cdots, m$ is an individual MPR.

In a similar manner, the group consensus index of each MPR, i.e., $GCI_{A^c}(A_l)$, $l = 1, 2, \cdots, m$, is based on the compatibility of $A_l$ w.r.t. the group opinion given by $A^c$ in Equation (3). Thus, the assessment of group consensus for each MPR $A_l$ given by $GCI_{A^c}(A_l)$ read as:

$$GCI_{A^c}(A_l) = d(A_l, A^c) \leq \overline{GCI}, \tag{9}$$

where the index is usually set at $\overline{GCI} = 1.1$ and $A_l$ is an individual MPR.

From Equations (7) and (8), for $CI_K$, it follows

$$\begin{aligned} &d(A_l, K) \leq \overline{CI} \Rightarrow \\ &1.0 \leq \frac{1}{n} \left[ \frac{1}{n} \Sigma_{i=1}^{n-1} \Sigma_{j=i+1}^n \left( a_{ij} \prod_{k=1}^n (a_{ik} a_{kj}) + a_{ji} \prod_{k=1}^n (a_{jk} a_{ki}) \right) + 1 \right] \leq \overline{CI}. \end{aligned} \tag{10}$$

Respectively from Equations (7) and (9), for $GCI_{A^c}$ evaluated for an $A_l$ follows:

$$d(A_l, A^c) \leq \overline{GCI} \Rightarrow 1.0 \leq \frac{1}{n} \left[ \frac{1}{n} \Sigma_{i=1}^{n-1} \Sigma_{j=i+1}^n \left( a_{ij,l} a_{ji}^c + a_{ji,l} a_{ij}^c \right) + 1 \right] \leq \overline{GCI}. \tag{11}$$

An MPR $A_t$ is completely consistent if and only if $CI_K(A_t) = 1$. Thus, a threshold useful to measure the similarity of two MPRs up to an acceptable level of consistency was suggested by [41,42] as $\overline{CI} = 1.1$. Similarly, an MPR $A_t$ is completely in consensus if and only if $GCI_{A^c}(A_t) = 1$ and an acceptable level of consensus is $\overline{GCI} = 1.1$.

In the following up to the end of the section, we utilize Definitions and Theorems recently introduced to measure the Individual Consistency Index and Group Consensus Index of a set of I-MPRs (cf. [43,44]). For details and proofs, please refer to these works.

**Definition 1.** *Let $\{ \overset{o}{A}_t \}_{n \times n} = \{ a_{ij}^{(t)} \}_{n \times n}$ be the set of MPRs generated by the combinations of $\bar{a}_{ij}^{(t)}$ and $\overset{+}{a}_{ij}^{(t)}$ entries of $A_t$ given by Equation (5), where $o = 1, 2, \cdots, \mu$ and $\mu = 2^{\frac{n(n-1)}{2}}$.*

**Definition 2.** *The Individual Consistency Index of the I-MPRs $(A_t)_{n \times n}$ given by Equation (4) when one has generated the set of MPRs $\{ \overset{o}{A}_t \}_{n \times n}$ given by Definition 1, is defined by*

$$CI_{\overset{*}{K}}(\overset{*}{A}_t) \equiv max\{ CI_{\underset{K}{1}}(\overset{1}{A}_t), CI_{\underset{K}{2}}(\overset{2}{A}_t), \cdots, CI_{\underset{K}{\mu-1}}(\overset{\mu-1}{A}_t), CI_{\underset{K}{\mu}}(\overset{\mu}{A}_t) \}. \tag{12}$$

**Definition 3.** *The smallest Individual Consistency Index of the I-MPRs $(A_t)_{n \times n}$ given by Equation (4) when one has generated the set of MPR $\{\overset{o}{A}_t\}_{n \times n}$ given by Definition 1, is defined by*

$$CI_{\tilde{K}}(\tilde{A}_t) \equiv min\{CI_{\underset{K}{1}}(\overset{1}{A}_t), CI_{\underset{K}{2}}(\overset{2}{A}_t), \cdots, CI_{\underset{K}{\mu-1}}(\overset{\mu-1}{A}_t), CI_{\underset{K}{\mu}}(\overset{\mu}{A}_t)\}. \tag{13}$$

From Definition 2, the next theorem is set forth.

**Theorem 1.** *Let $(A_t)_{n \times n} = (a_{ij}^{(t)})_{n \times n}$ be an I-MPR given by Equation (4) and generate the set of $\mu$ MPRs $\{\overset{o}{A}_t\}_{n \times n}$ by using the Definition 1. If one has an MPR, $(A_x)_{n \times n} = (a_{ij}^{(x)})_{n \times n}$ within the intervals $\left[\overset{-}{A}_t \ \overset{+}{A}_t\right]$ given by the I-MPRs $(A_t)_{n \times n}$, then*

$$CI_K(A_x) \leq CI_{\underset{K}{*}}(\overset{*}{A}_t). \tag{14}$$

In order to illustrate how useful are these Definitions and Theorem, we apply them on two I-MPRs given by two experts to assessing three criteria.

**Example 1.** *Let us consider two experts evaluating a set of three criteria through:*

$$A_1 = \begin{pmatrix} 1 & [1 \ 2] & [3 \ 4] \\ [1/2 \ 1] & 1 & [5 \ 6] \\ [1/4 \ 1/3] & [1/6 \ 1/5] & 1 \end{pmatrix}, \ A_2 = \begin{pmatrix} 1 & [1.1 \ 1.2] & [1.3 \ 1.4] \\ [1/1.2 \ 1/1.1] & 1 & [1.5 \ 1.6] \\ [1/1.4 \ 1/1.3] & [1/1.6 \ 1/1.5] & 1 \end{pmatrix}, \tag{15}$$

*where for $A_1$ it implies $w_1 \succ w_2 \succ w_3$, and for $A_2$ one gets $w_1 \succ w_2 \succ w_3$.*

*Note: Experts coincide in their rankings of the three criteria. Nevertheless, the assessment ratios are different.*

In the following, the Definitions 1–3 are figured out for these two I-MPRs.

From Definition 1 applied to $A_1$:

$$\overset{1}{A}_1 = \begin{pmatrix} 1 & 1 & 3 \\ 1 & 1 & 5 \\ 1/3 & 1/5 & 1 \end{pmatrix}, \ \overset{2}{A}_1 = \begin{pmatrix} 1 & 1 & 3 \\ 1 & 1 & 6 \\ 1/3 & 1/6 & 1 \end{pmatrix}, \overset{3}{A}_1 = \begin{pmatrix} 1 & 1 & 4 \\ 1 & 1 & 5 \\ 1/4 & 1/5 & 1 \end{pmatrix}, \cdots,$$

$$\overset{8}{A}_1 = \begin{pmatrix} 1 & 2 & 4 \\ 1/2 & 1 & 6 \\ 1/4 & 1/6 & 1 \end{pmatrix}. \tag{16}$$

In a similar manner, from Definition 1 applied to $A_2$:

$$\overset{1}{A}_2 = \begin{pmatrix} 1 & 1.1 & 1.3 \\ * & 1 & 1.5 \\ * & * & 1 \end{pmatrix}, \ \overset{2}{A}_2 = \begin{pmatrix} 1 & 1.1 & 1.3 \\ * & 1 & 1.6 \\ * & * & 1 \end{pmatrix}, \overset{3}{A}_2 = \begin{pmatrix} 1 & 1.1 & 1.4 \\ * & 1 & 1.5 \\ * & * & 1 \end{pmatrix}, \cdots,$$

$$\overset{8}{A}_2 = \begin{pmatrix} 1 & 1.2 & 1.4 \\ * & 1 & 1.6 \\ * & * & 1 \end{pmatrix}, \tag{17}$$

where, from now on, symbol $*$ corresponds to the inverted entries, respectively.

From Definition 2 applied to $A_1$ and $A_2$, it read as:

$$CI_{\underset{K}{*}}(\overset{*}{A}_1) = 1.072454, \ CI_{\tilde{K}}(\tilde{A}_1) = 1.0018; \ CI_{\underset{K}{*}}(\overset{*}{A}_2) = 1.005640, \ CI_{\tilde{K}}(\tilde{A}_2) = 1.0010. \tag{18}$$

*The first expert has the higher Individual Consistency Index which means that s/he has a weaker consistency in her/his judgments. It coincides with our logical expectations by analyzing and comparing both I-MPRs.*

*By defining for $A_1$ and $A_2$ an MPR within their own intervals, for instance*

$$A_{1x} = \begin{pmatrix} 1 & 1.5 & 3.7 \\ * & 1 & 5.6 \\ * & * & 1 \end{pmatrix}, \quad A_{2x} = \begin{pmatrix} 1 & 1.15 & 1.37 \\ * & 1 & 1.56 \\ * & * & 1 \end{pmatrix}, \tag{19}$$

*and by applying Theorem 1 to both MPR, one has:*

$$CI_{K_{1x}}(A_{1x}) = 1.0251 \leq CI_{\underset{K}{*}}(\overset{*}{A_1}); \quad CI_{K_{2x}}(A_{2x}) = 1.0027 \leq CI_{\underset{K}{*}}(\overset{*}{A_2}). \tag{20}$$

In order to reach a Group Consensus solution, the experts are supposed to participate in rounds of discussion to meet a common goal. Then, definitions are next introduced to compute the dissimilarity amongst DM' assessments for finally obtaining the GCI level.

In a similar manner as Definition 1, one can compute the whole combination of the values given by the set of I-MPRs through:

**Definition 4.** *From the m I-MPRs given as in Equation (4), $(A_t)_{n\times n}$, $t = 1, 2, \cdots, m$, their set of MPRs $\{\check{A}\}_{n\times n}$ which is the whole combination of the interval values of the set of $(A_t)_{n\times n}$ I-MPRs with $\overset{-(t)}{a_{ij}}$ and $\overset{+(t)}{a_{ij}}$, where $r = 1, 2, \cdots, v$ and $v = 2^{m\cdot\mu} = 2^{\frac{m(n)(n-1)}{2}}$, is given by*

$$\{\check{A}\}_{n\times n} = \{\overset{1}{\check{A}}, \overset{2}{\check{A}}, \cdots, \overset{v}{\check{A}}\}. \tag{21}$$

Furthermore, a set of $\{\check{A}^c\}_{n\times n}$ which represents each group opinion utilizing the geometric average operator corresponding to each element of the set $\{\check{A}\}_{n\times n}$, can be defined.

**Definition 5.** *Based on Definition 4, one can compute for the set of I-MPRs $(A_t)_{n\times n}$ and $\{\check{A}\}_{n\times n}$, $t = 1, 2, \cdots, m$, its set of MPR $\{\check{A}^c\}_{n\times n}$. Then, the corresponding set of $\{\check{A}^c\}_{n\times n}$ which is the Group I-MPRs Opinion utilizing the geometric operator is given by*

$$\{\check{A}^c\}_{n\times n} = \{\check{a}^c_{ij}\}_{n\times n} = \{\prod_{t=1}^{m}(\check{A})^{\lambda_t}\}, \quad t \in M, \tag{22}$$

*where $\{\check{a}^c_{ij}\}_{n\times n}$ is the combination of the whole set of values ($2^{m\cdot\mu} = 2^{\frac{m\cdot n\cdot(n-1)}{2}}$).*

**Definition 6.** *The Group Consensus Index of a set of I-MPRs $(A_t)_{n\times n}$, $t = 1, 2, \cdots, m$, given by Equation (4) when one has generated the set of MPRs $\{\check{A}^c\}_{n\times n} = \{\overset{1}{\check{A}^c}, \overset{2}{\check{A}^c}, \cdots, \overset{v}{\check{A}^c}\}$, given by Definition 5, is defined by*

$$GCI_{\check{H}_t}(\check{A}_t)_{n\times n} \equiv \max\{GCI_{\underset{\check{A}^c}{1}}(\overset{1}{\check{A}}), GCI_{\underset{\check{A}^c}{2}}(\overset{2}{\check{A}}), \cdots, GCI_{v-1}(\overset{v-1}{\check{A}}), GCI_{\underset{\check{A}^c}{v}}(\overset{v}{\check{A}})\}, \tag{23}$$

*where $v = 2^{\frac{m\cdot n\cdot(n-1)}{2}}$ and $\check{H}_t$ is associated with the corresponding $\overset{p}{\check{A}^c}$ matrix and $p \in \{1, 2, \cdots, v\}$.*

**Definition 7.** *The smallest Group Consensus Index of a set of I-MPRs $(A_t)_{n\times n}$, $t = 1, 2, \cdots, m$, given by Equation (4) when one has generated the set of MPRs $\{\check{A}^c\}_{n\times n} = \{\overset{1}{\check{A}^c}, \overset{2}{\check{A}^c}, \cdots, \overset{v}{\check{A}^c}\}$, given by Definition 5, is defined by*

$$GCI_{\overline{H}_t}\left(\overset{-}{A}_t\right)_{n\times n} \equiv \min\{GCI_{\underset{\check{A}^c}{1}}(\overset{1}{\check{A}}), GCI_{\underset{\check{A}^c}{2}}(\overset{2}{\check{A}}), \cdots, GCI_{\underset{\check{A}^c}{\nu-1}}(\overset{\nu-1}{\check{A}}), GCI_{\underset{\check{A}^c}{\nu}}(\overset{\nu}{\check{A}})\}, \tag{24}$$

*where $\nu = 2^{\frac{m\cdot n\cdot(n-1)}{2}}$ and $\overline{H}_t$ is associated with the corresponding $\overset{p}{\check{A}^c}{}_{n\times n}$ matrix and $p \in \{1, 2, \cdots, \nu\}$.*

**Example 2.** *Let us consider the confidence weight of the two experts through the following weight vector $\lambda = [2/7 \; 5/7]$. Then, from Definition 4, Equations (16) and (17), it follows:*

$$\overset{1}{\check{A}} = \{\overset{1}{A}_1, \overset{1}{A}_2\}; \overset{2}{\check{A}} = \{\overset{1}{A}_1, \overset{2}{A}_2\}; \overset{3}{\check{A}} = \{\overset{1}{A}_1, \overset{3}{A}_2\}; \cdots; \overset{64}{\check{A}} = \{\overset{8}{A}_1, \overset{8}{A}_2\}. \tag{25}$$

*From Definition 5 applied to Equation (25), one has:*

$$\{\check{A}^c\} = \{\overset{1}{\check{A}^c}, \overset{2}{\check{A}^c}, \cdots, \overset{64}{\check{A}^c}\}, \tag{26}$$

*where, from Equation (3), it implies:*

$$
\begin{aligned}
\overset{1}{\check{A}^c} &= (\overset{1}{\check{a}^c}_{ij}) = \left(\overset{1(1)}{a}_{ij}\right)^{2/7} * \left(\overset{1(2)}{a}_{ij}\right)^{5/7}; \quad \overset{2}{\check{A}^c} = (\overset{2}{\check{a}^c}_{ij}) = \left(\overset{1(1)}{a}_{ij}\right)^{2/7} * \left(\overset{2(2)}{a}_{ij}\right)^{5/7}; \\
\overset{3}{\check{A}^c} &= (\overset{3}{\check{a}^c}_{ij}) = \left(\overset{1(1)}{a}_{ij}\right)^{2/7} * \left(\overset{3(2)}{a}_{ij}\right)^{5/7}; \cdots; \overset{64}{\check{A}^c} = (\overset{64}{\check{a}^c}_{ij}) = \left(\overset{8(1)}{a}_{ij}\right)^{2/7} * \left(\overset{8(2)}{a}_{ij}\right)^{5/7}.
\end{aligned} \tag{27}
$$

*From Definition 6, it results:*

$$GCI_{\check{H}_1}(\check{A}_1) = 1.2143; \quad GCI_{\check{H}_2}(\check{A}_2) = 1.0325. \tag{28}$$

*These results indicate that the first expert $A_1$, which has the lowest weight ($\lambda_1 = 2/7$), is not in consensus. In addition, from Definition 7, it follows:*

$$GCI_{\overline{H}_1}\left(\overline{A}_1\right) = 1.1120; \quad GCI_{\overline{H}_2}\left(\overline{A}_2\right) = 1.0173. \tag{29}$$

As soon as we obtain ICI and GCI of a set of I-MPRs, the accepted threshold values ($\overline{ICI}$ and $\overline{GCI}$ respectively) give how acceptable in Consistency and Consensus the assessments of the DMs are. Whenever some I-MPRs are over their index values, we can apply the following strategy to improve their Individual Consistency and/or their Group Consensus Indices.

## 3. Reliable Intervals for Individual Consistency and Group Consensus

Once the ICI has been calculated, it may happen that one or more I-MPRs are not consistent. For example, $r$ I-MPRs were not consistent. In that case, we need to improve the Individual Consistency of those I-MPRs and then let us define this set of MPRs as:

$$A_I^o \equiv \left[\overset{-(o)}{a}_{ij} \quad \overset{+(o)}{a}_{ij}\right]_{n\times n}, \quad i, j \in N, \; o = 1, 2, \cdots, r. \tag{30}$$

Naturally, each one of the $A_I^o$, $I = 1, 2, \cdots, r$ doesn't verify for a special combination of its values (given by Equation (12)), the inequality given in Equation (10).

In a similar manner, once the GCI has been calculated, it may happen that one or more I-MPRs are not in consensus. For example, for instance $y$ I-MPRs were not in consensus. In that case, we need to improve the Group Consensus of those I-MPRs and then let us define this set of MPRs as

$$A_J^p \equiv \begin{bmatrix} \overline{a}_{ij}^{(p)} & \overset{+}{a}_{ij}^{(p)} \end{bmatrix}_{n \times n}, \quad i,j \in N, \ p = 1,2,\cdots,y. \tag{31}$$

Once again, each one of the $A_J^P$, $J = 1, 2, \cdots, y$ doesn't verify, for a special combination of values (given by Equation (23)), the inequality given in Equation (11).

### 3.1. Sequential Quadratic Programming Methodology

When one has a constrained nonlinear optimization problem (NLP) and wants a numerical solution, the Sequential Quadratic Programming (SQP) [45] is a useful approach. Let us consider the implementation of the sequential quadratic programming methodology to NLP of the form:

$$\begin{aligned} \text{minimize} \quad & f(x), \\ \text{over } x \quad & \in \quad \mathbb{R}^n, \\ \text{subject to } h(x) \quad & = \quad 0, \\ g(x) \quad & \leq \quad 0, \end{aligned} \tag{32}$$

where $f : \mathbb{R}^n \longrightarrow \mathbb{R}$ is the objective functional, the functions $h : \mathbb{R}^n \longrightarrow \mathbb{R}^m$ and $g : \mathbb{R}^n \longrightarrow \mathbb{R}^p$ describe the equality and inequality constraints. The sequential quadratic programming is an iterative method for modelling the NLP for a given iterate $x^k$, $k \in \mathbb{N}_0$, by a Quadratic Programming (QP) subproblem. As soon as the method solves that QP subproblem, the solution is used to build a new iterate $x^{k+1}$. This is figured out in such a way that the sequence $(x^k)_{k \in \mathbb{N}_0}$ converges to a local minimum $x^*$ of the NLP Equation (32) as $k \to \infty$.

Then, as we can observe, the problem of minimizing an objective functional (given in our case by Equation (7)), where one has some inequalities constraints from Equation (10), (related to $g(x)$) with the additional observation of constraints imposed by Equation (30) (related to $h(x)$), defines an NLP to improve the Individual Consistency Index of an I-MPR.

In a similar manner, the problem of minimizing an objective functional (given by Equation (7)) providing some inequalities constraints from Equation (11), (related to $g(x)$) with the additional observation of constraints imposed by Equation (31) (related to $h(x)$), defines an NLP to improve the Group Consensus Index of a set of I-MPRs.

An additional observation is that since we have interval judgments in I-MPRs, we should apply the SQP to minimize the objective functional to found the minimal values verifying the required constraints. After that, we should then apply the SQP to maximize the objective functional to found the maximal values. In this manner, we will finally find reliable intervals where are fulfilled inequalities and equalities constraints and conditions of Equations (10) and (11), to obtain acceptable Individual Consistency and Group Consensus, respectively.

### 3.2. Matching the Problem with the SQP for Improving I-MPRs

From Equation (30) (Equation (31) resp.), the following inequalities can be stated in terms of optimization variables $x_i$, as follows:

$$\begin{aligned} \overline{a}_{12}^{\delta} \quad & \leq \quad x_1 \quad \leq \quad \overset{+\delta}{a}_{12}, \\ & \vdots \\ \overline{a}_{1n}^{\delta} \quad & \leq \quad x_{n-1} \quad \leq \quad \overset{+\delta}{a}_{1n}, \\ & \vdots \\ \overline{a}_{(n-1)n}^{\delta} \quad & \leq \quad x_{\frac{n^2-n}{2}} \quad \leq \quad \overset{+\delta}{a}_{(n-1)n}, \end{aligned} \tag{33}$$

where $\delta$ stands for $o$ in the Individual Consistency assessments and for $p$ in the Group Consensus assessments. For the Individual Consistency, the inequalities given from Equation (10) are

$1.0 \leq d(\overset{*}{A}_I, \overset{*}{K}) \leq 1.1$ which impose some inequality constraints to be fulfilled. For Group Consensus improvement, they are given from Equation (11) as $1.0 \leq d(\check{A}_J, \check{H}_J) \leq 1.1$.

The relationship between $x_k$, $k = 1, 2, \cdots, \frac{n^2-n}{2}$ and the set of I-MPRs $(A_t)_{n \times n}$, $t = 1, 2, \cdots, m$ under analysis is given by:

$$
A_t =
\begin{pmatrix}
1 & \overset{-+(t)}{a}_{12} & \overset{-+(t)}{a}_{13} & \cdots & \overset{-+(t)}{a}_{1(n-1)} & \overset{-+(t)}{a}_{1n} \\
* & 1 & \overset{-+(t)}{a}_{23} & \cdots & \overset{-+(t)}{a}_{2(n-1)} & \overset{-+(t)}{a}_{2n} \\
* & * & 1 & \cdots & \overset{-+(l)}{a}_{3(n-1)} & \overset{-+(t)}{a}_{3n} \\
\vdots & \cdots & \cdots & \ddots & \vdots & \vdots \\
* & * & \cdots & \cdots & 1 & \overset{-+(t)}{a}_{(n-1)n} \\
* & * & \cdots & \cdots & * & 1
\end{pmatrix}
\equiv
$$

$$
X_t =
\begin{pmatrix}
1 & x_{1+\frac{n(n-1)}{2}(t-1)} & x_{2+\frac{n(n-1)}{2}(t-1)} & \cdots & x_{n-1+\frac{n(n-1)}{2}(t-1)} \\
* & 1 & x_{n+\frac{n(n-1)}{2}(t-1)} & \cdots & x_{2(n-1)-1+\frac{n(n-1)}{2}(t-1)} \\
* & * & 1 & \cdots & x_{3(n-1)-3+\frac{n(n-1)}{2}(t-1)} \\
\vdots & \cdots & \cdots & \ddots & \vdots \\
* & * & \cdots & \cdots & x_{\frac{n(n-1)}{2}t} \\
* & * & \cdots & \cdots & 1
\end{pmatrix},
$$

(34)

where $\overset{-+(t)}{a}_{ij}$ stands for the corresponding crisp or interval value of the I-MPR.

From the definition of constrained nonlinear optimization problem, we note that when the I-MPR is in fact just an MPR (when the respective expert has a high confidence in her/his assessments), the SQP could not give any different solution. This is because the optimization variables will remain unchanged since the variables $x_i$, $i = 1, 2, 3, \cdots, (n^2 - n)/2$ will be defined as a crisp values (cf. Equation (33)).

Thus, in order to provide a general benchmark to address and solve any possible case of assessments given through I-MPRs, we will modify slightly the SQP. We introduce a design parameter $\epsilon$ which will be used as an additive or subtractive element, for high and low bounds, respectively, of the I-MPRs to be improved.

For example, we modify Equation (33) as follows:

$$
\begin{aligned}
\bar{a}^\delta_{12} - \epsilon & \leq & x_1 & \leq & \overset{+\delta}{a}_{12} + \epsilon, \\
& & \vdots & & \\
\bar{a}^\delta_{1n} - \epsilon & \leq & x_{n-1} & \leq & \overset{+\delta}{a}_{1n} + \epsilon, \\
& & \vdots & & \\
\bar{a}^\delta_{(n-1)n} - \epsilon & \leq & x_{\frac{n^2-n}{2}} & \leq & \overset{+\delta}{a}_{(n-1)n} + \epsilon,
\end{aligned}
$$

(35)

where $\epsilon > 0$. For the sake of compactness, let us integrate in the same terms the parameter $\epsilon$. For example, $\bar{a}^{(o)}_{ij} \equiv \bar{a}^o_{ij} - \epsilon$ and $\overset{+(o)}{a}_{ij} \equiv \overset{+o}{a}_{ij} + \epsilon$ for Individual Consistency analysis, and $\bar{a}^{(p)}_{ij} \equiv \bar{a}^p_{ij} - \epsilon$ and $\overset{+(p)}{a}_{ij} \equiv \overset{+p}{a}_{ij} + \epsilon$, for Group Consensus analysis.

In that manner, when the SQP does not provide a feasible solution, we will again iterate with an incremental $\epsilon$-value, until a solution is found.

It is worth mentioning that $\epsilon$ must be initialized, with a small value, to keep the maximum of the information provided by the expert.

### 3.2.1. Individual Consistency Objective Functional

Thus, when an I-MPR is inconsistent, we need to synthesize the objective functional. For example, from Equations (7) and (10), one obtains:

$$1.0 \leq \frac{1}{n}\left[\frac{1}{n}\Sigma_{i=1}^{n-1}\Sigma_{j=i+1}^{n}\left(a_{ij}\prod_{k=1}^{n}(a_{ik}a_{kj}) + a_{ji}\prod_{k=1}^{n}(a_{jk}a_{ki})\right) + 1\right] \leq \overline{CI}. \tag{36}$$

After some algebraic manipulations, it implies:

$$1.0 \leq \frac{1}{n^2}\left[\Sigma_{i=1}^{n-1}\Sigma_{j=i+1}^{n}\left(a_{ij}^{\frac{n-2}{n}}\prod_{\substack{l=1\\i\neq l,l\neq j}}^{n}(a_{jl}a_{li}) + a_{ji}^{\frac{n-2}{n}}\prod_{\substack{l=1\\i\neq l,l\neq j}}^{n}(a_{il}a_{lj})\right)\right] + \frac{1}{n} \leq \overline{CI}. \tag{37}$$

### 3.2.2. Group Consensus Objective Functional

In a similar manner, when an I-MPR $(A_J^o)_{n\times n}$ is not in consensus, we need to synthesize the objective functional. For example, from Equation (11), one obtains:

$$1.0 \leq \frac{1}{n^2}\left[\Sigma_{i=1}^{n-1}\Sigma_{j=i+1}^{n}\left(a_{ij}^{(J)}\prod_{t=1}^{m}(a_{ji}^{(t)})^{\lambda_t} + a_{ji}^{(J)}\prod_{t=1}^{m}(a_{ij}^{(t)})^{\lambda_t}\right)\right] + \frac{1}{n} \leq \overline{GCI}. \tag{38}$$

After some algebraic manipulations, it implies for $(A_J)_{n\times n}$:

$$1.0 \leq \frac{1}{n^2}\left(\underbrace{\frac{a_{12}^{(J)(1-\lambda_J)}}{a_{12}^{(1)(\lambda_1)}\cdot a_{12}^{(2)(\lambda_2)}\cdots a_{12}^{(m)(\lambda_m)}}}_{} + \frac{1}{\alpha_1} + \underbrace{\frac{a_{13}^{(J)(1-\lambda_J)}}{a_{13}^{(1)(\lambda_1)}\cdot a_{13}^{(2)(\lambda_2)}\cdots a_{13}^{(m)(\lambda_m)}}}_{} + \frac{1}{\alpha_2} + \right.$$
$$\left. \cdots + \underbrace{\frac{a_{(n-1)n}^{(J)(1-\lambda_J)}}{a_{(n-1)n}^{(1)(\lambda_1)}\cdot a_{(n-1)n}^{(2)(\lambda_2)}\cdots a_{(n-1)n}^{(m)(\lambda_m)}}}_{} + \frac{1}{\alpha_m}\right) + \frac{1}{n} \leq \overline{GCI}, \tag{39}$$

where $\alpha_1$ equals to the term over the first brace, $\alpha_2$ equals to the term over the second brace, and so on. Furthermore, note that the term $a_{12}^{(1)(\lambda_1)}\cdot a_{12}^{(2)(\lambda_2)}\cdots a_{12}^{(m)(\lambda_m)}$ excepts the term $a_{12}^{(J)(\lambda_J)}$, the term $a_{13}^{(1)(\lambda_1)}\cdot a_{13}^{(2)(\lambda_2)}\cdots a_{13}^{(m)(\lambda_m)}$ excepts the term $a_{13}^{(J)(\lambda_J)}$, and so on.

Finally, the initialization point $x_0$ used in the NLP can be set at any point within the corresponding interval given by Equation (33).

In the following, an algorithm based on an NLP is used to obtain reliable intervals for the assessment of based decision models such as those given by the set of I-MPRs for both indices ($\overline{CI}$ and $\overline{GCI}$). Two similar methods are blended to obtain both reliable I-MPR Consistency and Consensus levels. For example, for Individual Consistency improvement, we can find an I-MPR verifying Individual Consistency. For the other case, for Group Consensus improvement, a reliable Consensus Index for I-MPRs is obtained. In order to use an NLP, an SQP algorithm can be found in [45]. In the following, our algorithm is described in detail.

### 3.3. Improving the Individual Consistency of an I-MPR

A scheme of the Algorithm 1 implementation is depicted in Figure 1. Once the set of I-MPRs are all acceptably consistent, we can improve the Group Consensus level through the following algorithm.

---

**Algorithm 1:** Algorithm IC-I-MPR

---

**Input:** $A_I^o = (\overset{-(o)}{a_{ij}} \;\; \overset{+(o)}{a_{ij}})_{n \times n}$: the initial interval I-MPR; $x_0$: the initialization value for the nonlinear optimization, which is to be defined within the corresponding interval or as the corresponding crisp value; the threshold value of $\overline{CI}$ for the Individual Consistency assessment; $\epsilon_c$ : Design parameter value allowing the enlargement pace of the searching space of the algorithm.

**Output:** $\overline{A}_I^o$: the consistency interval matrix computed and verifying interval conditions given by Equation (10).

**Step 1:** Get the function for the assessment of Individual Consistency given by Equation (37).

**Step 2:** Define for the nonlinear optimization algorithm, the allowed intervals:

$$\overset{-(o)}{a_{ij}} \leq x_i \leq \overset{+(o)}{a_{ij}} \; ; j > i, \;\; i, j = 1, 2, \cdots, n. \tag{40}$$

Thus, assign the former linear inequality constraints as follows:

$$
\begin{aligned}
c(2k+1) &= x_i - \overset{+(o)}{a_{ij}} \; ; \;\; c(2(k+1)) = \overset{-(o)}{a_{ij}} - x_i, \\
j > i, \;\; i, j &= 1, 2, \cdots, n, \quad k = 0, 1, 2, \cdots, \tfrac{n^2 - n}{2}.
\end{aligned}
\tag{41}
$$

**Step 3:** Obtain the acceptable index of Individual Consistency, $1.0 \leq d(\overset{*}{A}_I, \overset{*}{K}) \leq \overline{CI}$. Thus, based on matrix $K$ given by Equation (2), the nonlinear inequality constraints imposed are given by Equation (10).

**Step 4:** Solve the former nonlinear optimization problem using the SQP algorithm to minimize it.

**Step 5:** If an unfeasible solution is obtained, assign $\epsilon_c = \beta * \epsilon_c$, where $\beta = 1, 2, 3, \cdots$, increments at each iteration and return to **Step 4**. Otherwise, continue to the next step.

**Step 6:** Obtain $\overline{A}_{Imin}^o = (\overset{-(o)}{a_{ij}})_{n \times n}$. Solve again the same nonlinear optimization problem but this time in order to maximize it. In order to do so, assign the objective functional as $-f(x)$. Obtain $\overline{A}_{Imax}^o = (\overset{+(o)}{a_{ij}})_{n \times n}$.

**Step 7:** Compose the Consistency Interval Matrix $\overline{A}_I^o$ as follows:

$$\overline{A}_I^o = \begin{bmatrix} \overset{-(o)}{a_{ij}} & \overset{+(o)}{a_{ij}} \end{bmatrix}_{n \times n}, \tag{42}$$

where $\begin{bmatrix} \overset{-(o)}{a_{ij}} & \overset{+(o)}{a_{ij}} \end{bmatrix}$, stands for the interval or crisp value obtained.

**Step 8:** end.

---

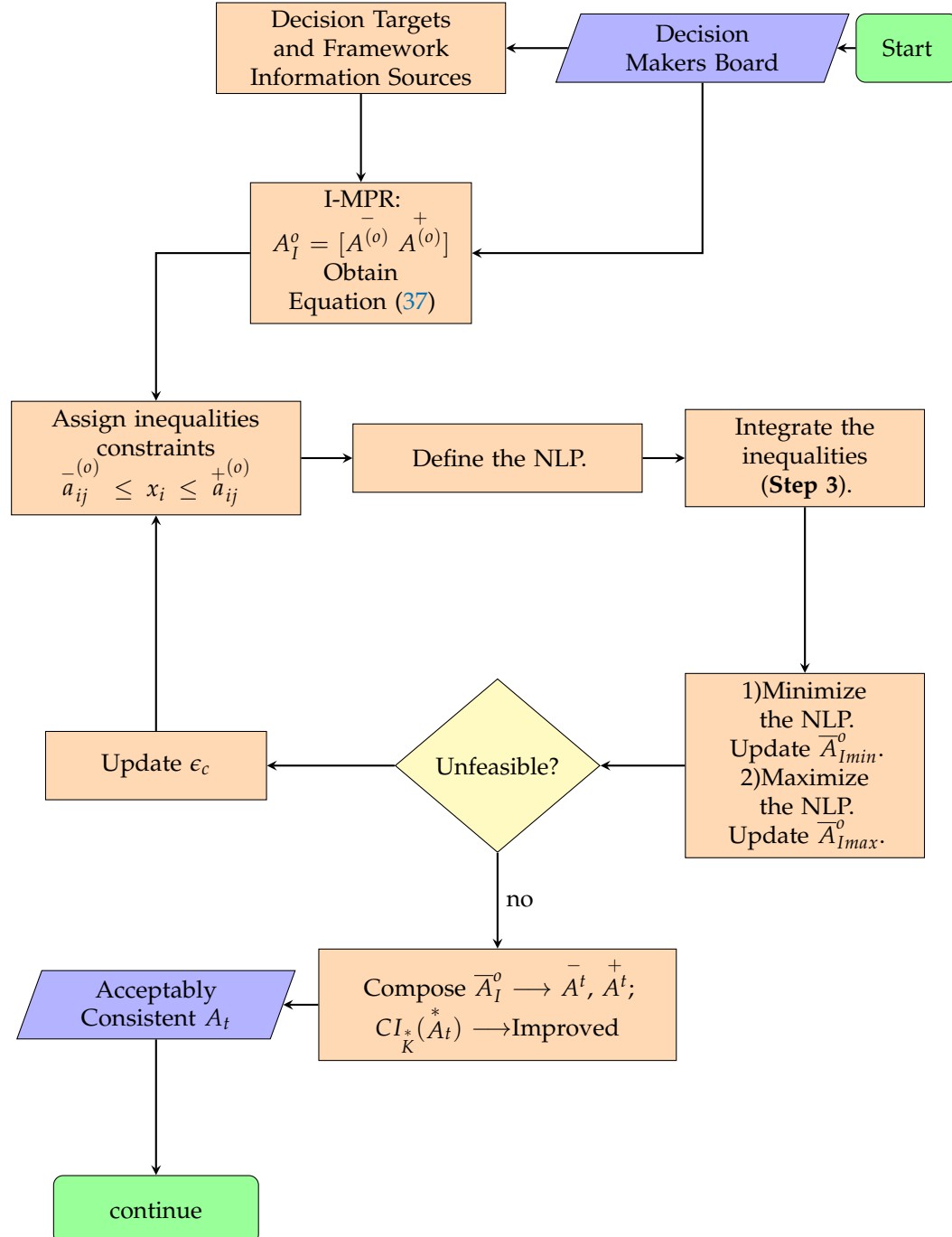

**Figure 1.** Process flowchart for improving consistency of an I-MPR.

### 3.4. Improving the Group Consensus of a Set of I-MPRs

A scheme of the Algorithm 2 implementation is depicted in Figure 2.

---

**Algorithm 2:** Algorithm GC-I-MPR

---

**Input:** $A_J^P = (\overset{-(p)}{a}_{ij} \overset{+(p)}{a}_{ij})_{n \times n}$: the initial interval I-MPRs; $x_0$: the initialization value for the nonlinear optimization, which is to be defined within the corresponding interval or as the corresponding crisp value; $\overline{GCI}$ for the Group consensus assessment; $\epsilon_g$ : design parameter allowing the enlargement of the searching space of the algorithm.

**Output:** $\overline{A}_J^P$: the I-MPRs computed and verifying interval conditions given by Equation (11).

**Step 1:** Get the function for the assessment of Group Consensus given by Equation (39).

**Step 2:** Define for the nonlinear optimization algorithm, the allowed intervals:

$$\overset{-(p)}{a}_{ij} \leq x_i \leq \overset{+(p)}{a}_{ij} \; ; j > i, \; i, j = 1, 2, \cdots, n. \tag{43}$$

Thus, assign the former linear inequality constraints as follows:

$$
\begin{aligned}
c(2k+1) &= x_i - \overset{+(p)}{a}_{ij} \; ; \; c(2(k+1)) = \overset{-(p)}{a}_{ij} - x_i; \\
j > i, \; i, j &= 1, 2, \cdots, n, \quad k = 0, 1, 2, \cdots, \frac{n^2-n}{2}.
\end{aligned}
\tag{44}
$$

**Step 3:** Obtain the acceptable Group Consensus Index $d(\check{A}_J, \check{H}_J)$. Thus, based on the set of $\{\check{A}^c\}_{n \times n}$ given by Equation (22), the nonlinear inequality constraints imposed is given by $1.0 \leq d(\check{A}_J, \check{H}_J) \leq 1.1$.

**Step 4:** If $GCI(A_t) \leq \overline{GCI}, t = 1, 2, \cdots, m$, then goto **Step 9**. Otherwise, continue with the next step.

**Step 5:** Solve the former nonlinear optimization problem (NLP) using the SQP algorithm to minimize it.

**Step 6:** If an unfeasible solution is obtained, assign $\epsilon_g = \theta * \epsilon$, where $\theta = 1, 2, 3, \cdots$, increments at each iteration and return to **Step 4**. Otherwise, continue to the next step.

**Step 7:** Obtain the matrix $\overline{A}_{Jmin}^P = (\overset{-(p)}{a}_{ij})_{n \times n}$. Solve again the same nonlinear optimization problem but this time in order to maximize it. Obtain $\overline{A}_{Jmax}^P = (\overset{+(p)}{a}_{ij})_{n \times n}$.

**Step 8:** Goto to **Step 2**.

**Step 9:** Compose the $J - th$ Group Consensus Interval Matrix $\overline{A}_J^P$ as follows:

$$\overline{A}_J^P = \left[ \overset{-(p)}{a}_{ij} \overset{+(p)}{a}_{ij} \right]_{n \times n}, \tag{45}$$

where $\left[ \overset{-(p)}{a}_{ij} \overset{+(p)}{a}_{ij} \right]$, stands for the interval or crisp value obtained.

**Step 10:** end.

---

Section Remarks:

- In the case that an expert has provided a crisp value(s) in her/his judgement(s), this value(s) drives the process of nonlinear optimization since they will slightly change with the pace of $\epsilon$. It is very useful since precisely in that value(s), the expert has shown her/his highest confidence level.
- At the end of both algorithms, one gets reliable I-MPRs, i.e., where the consistency and consensus constraints are fulfilled.

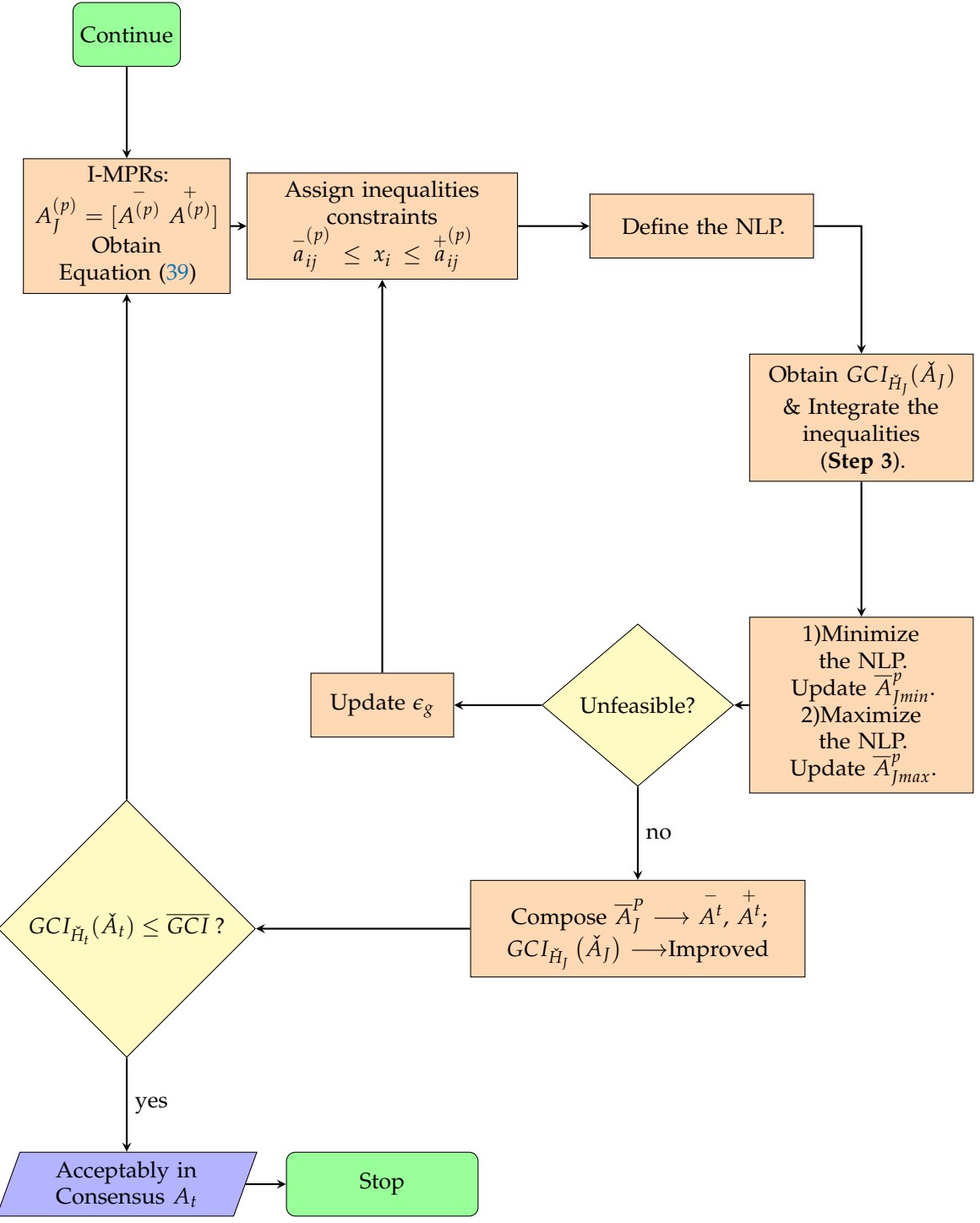

**Figure 2.** Process flowchart for improving consensus of I-MPRs.

## 4. Prioritization Method and Methodology Application

The process of deriving a priority weight vector of the alternatives, $w_i = (w_1, w_2, \cdots, w_n)^T$ from an I-MPR, for instance an $A_t$, is called a prioritization method, where $w_l \geq 0$ and $\sum_{l=1}^n w_l = 1$. Then,

when one has the evaluation for $\check{A}$ which is associated to the maximum value (cf. Equation (23)) of the Group Consensus Index of I-MPRs $A_t$, it follows from Equation (9):

$$\check{w}_i = \left( \left( \prod_{j=1}^{n} \overset{*}{a}_{ij} \right)^{1/n} \right) \cdot \left( \left( \prod_{j=1}^{n} \check{H}_t \right)^{1/n} \right) = w_i(\check{A}) w_i(\check{H}_t), \tag{46}$$

where $w_i(\check{A})$ and $w_i(\check{H}_t)$ are the weights of $\check{A}$ and $\check{H}_t$, respectively.

In a similar manner, for $\overline{A}$ which is associated to the minimum value (cf. Equation (13)) of the Group Consensus Index of I-MPRs $A_t$, one has:

$$\overline{w}_i(\theta) = \left( \left( \prod_{j=1}^{n} \overline{a}_{ij} \right)^{1/n} \right) \cdot \left( \left( \prod_{j=1}^{n} \overline{H_t} \right)^{1/n} \right) = w_i(\overline{A}) w_i(\overline{H_t}), \tag{47}$$

where $w_i(\overline{A})$ and $w_i(\overline{H_t})$ are the weights of $\overline{A}$ and $\overline{H_t}$, respectively.

Consequently:

$$w_i = [\overline{w}_i, \ \check{w}_i]. \tag{48}$$

As soon as the set of I-MPRs is acceptably consistent and in consensus, the interval priority vector to rank alternatives is obtained. To do so, an interval ranking is used.

*Interval Priority Vector Synthesis*

An interval priority vector should reflect different expert's risk preferences for her/his interval judgments. There are mainly two prioritization methods: (among others cf. [46] and the references therein) Eigenvalue-based Methods (EM), (cf. [1,47]) and the Row Geometric Mean Method (RGMM), (cf. [48]) that are utilized to derive a priority weight vector from an ordered judgment matrix which is the method here utilized.

Based on the results given in [49], a slight modification of their method is addressed below.

Let us consider for instance two intervals $a = \left[ \overline{a}, \overset{+}{a} \right]$ and $b = \left[ \overline{b}, \overset{+}{b} \right]$ where $\overline{a}, \overline{b} > 0$ and $a, b$ are positioned on the $x$ and $y$ axis, respectively. A uniform probability distribution is assumed on the constrained area composed of $a$ and $b$. For upper left points on $y = x$, the $y$ values are larger than $x$ values and viceversa for the lower right points (cf. Figure 3 where one possible case is shown).

Then, the preference degree of $P(a > b)$ is equal to $\frac{S_1}{d(a)d(b)}$, where $d(a) = \overset{+}{a} - \overline{a}$ and $d(b) = \overset{+}{b} - \overline{b}$ and the following ranking interval method can be stated.

**Definition 8.**

$$P(a > b) = \begin{cases} 1 & \overline{a} \geq \overset{+}{b}, \\ 1 - \frac{(\overset{+}{b} - \overline{a})^2}{2d(a)d(b)} & \overline{b} \leq \overline{a} < \overset{+}{b} \leq \overset{+}{a}, \\ \frac{2\overset{+}{a} - (\overset{+}{b} + \overline{b})}{2d(a)} & \overline{a} < \overline{b} < \overset{+}{b} \leq \overset{+}{a}, \end{cases} \tag{49}$$

*where if $P(a > b) > 0.5$, then $a > b$; if $P(a > b) = 0.5$, then $a = b$ and if $P(a > b) < 0.5$, then $a < b$; and in this manner Equation (49) indicates the total order of intervals and their preference degrees [50].*

In order to provide access to the public as on-line security and resource conditions allow, these complete methodology (definitions, applied theorems, algorithms and interval priority vector synthesis) will be soon available in a site. Then, this benchmark will be useful to test a submitted set of I-MPRs for assessment on their Individual Consistency (ICI) and the Group Consensus Indices (GCI). This site will test different data on various based decision models.

In the following, our main methodology is applied to numerical examples under diverse considerations in order to test different situations when the system is working in a real scenario.

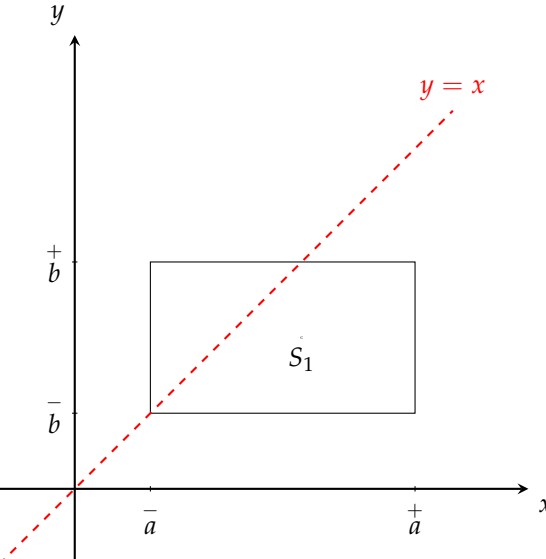

**Figure 3.** One case for the interval analysis.

## 5. Illustration of the Methodology through Numerical Examples

In the following example, let us suppose that three managers from different operating departments are participating in a group decision about the evaluation of a new market to have the best return of investment of a set of products. There are four decision criteria which involve the details of judgments, which are: development, legal restrictions for economic activities, society and infrastructure denoted as $C_1$, $C_2$, $C_3$, and $C_4$, respectively. The three mangers state their preferences over the four criteria and, since they are confident on the analysis carried out by their operating department, they have a crisp value for criteria judgments related to their department. Nevertheless, they provide an interval for others criteria assessments. Their associated weight vector is $\lambda = [1/3, 1/3, 1/3]$.

**Example 3.** *The three managers provide their I-MPRs through*

$$A_1 = \begin{pmatrix} 1 & [1\ 2] & 1/5 & [1/3\ 1/2] \\ * & 1 & 1/7 & [5\ 6] \\ * & * & 1 & 1/2 \\ * & * & * & 1 \end{pmatrix}, A_2 = \begin{pmatrix} 1 & 5/2 & 1/3 & 3/4 \\ * & 1 & [1/2\ 3/2] & [2\ 3] \\ * & * & 1 & [1\ 2] \\ * & * & * & 1 \end{pmatrix},$$

$$A_3 = \begin{pmatrix} 1 & 2 & [1/2\ 3/2] & [1\ 3/2] \\ * & 1 & 3/4 & 3 \\ * & * & 1 & [2\ 3] \\ * & * & * & 1 \end{pmatrix},$$

(50)

*where the initial ranking given by expert 1 is* $w_3 \overset{1}{>} w_2 \overset{0.3724}{>} w_4 \overset{1}{>} w_1$, *by expert 2 is:* $w_3 \overset{1}{>} w_2 \overset{0.967}{>} w_1 \overset{1}{>} w_4$, *and by expert 3 is:* $w_3 \overset{0.8179}{>} w_1 \overset{1}{>} w_2 \overset{1}{>} w_4$.

*Note that the first manager has more information and expertise on the third criteria. On the other hand, the second manager is focused on the first criteria and finally the third manager is focused on the second criteria.*

*Let us now apply Definition 1 to $A_1$, $A_2$ and $A_3$ as follows:*

$$\{\overset{o}{A_1}\} = \{\overset{1}{A_1}, \overset{2}{A_1}, \cdots, \overset{63}{A_1}, \overset{64}{A_1}\}; \ \{\overset{o}{A_2}\} = \{\overset{1}{A_2}, \overset{2}{A_2}, \cdots, \overset{63}{A_2}, \overset{64}{A_2}\}; \ \{\overset{o}{A_3}\} = \{\overset{1}{A_3}, \overset{2}{A_3}, \cdots, \overset{63}{A_3}, \overset{64}{A_3}\},$$

(51)

where $\{\overset{1}{A_r}, \overset{2}{A_r}, \cdots, \overset{64}{A_r}\}$, $r = 1, 2, 3$ are obtained in a similar manner as in Example *1*.

Then, $CI_{\overset{*}{K}}(\overset{*}{A_i})$, $i = 1, 2, 3$ is calculated through Equation (*2*) where, for the first expert, one obtains:

$$\overset{*}{A_1} = \begin{pmatrix} 1 & 2 & 1/5 & 1/3 \\ * & 1 & 1/7 & 6 \\ * & * & 1 & 1/2 \\ * & * & * & 1 \end{pmatrix}, \tag{52}$$

and its corresponding consistent MPR (calculated by Equation (*2*)) is

$$\overset{*}{K_1} = \begin{pmatrix} 1 & 0.7468 & 0.2954 & 0.6043 \\ * & 1 & 0.3956 & 0.8091 \\ * & * & 1 & 2.0453 \\ * & * & * & 1 \end{pmatrix}. \tag{53}$$

Similar calculations are carried out for $A_2$ and $A_3$.

Then, from Definition *3*, $CI_{\tilde{K}}(\tilde{A}_i)$, $i = 1, 2, 3$ is calculated through Equation (*13*) where for the first expert one obtains:

$$\tilde{A}_1 = \begin{pmatrix} 1 & 1 & 1/5 & 1/2 \\ * & 1 & 1/7 & 5 \\ * & * & 1 & 1/2 \\ * & * & * & 1 \end{pmatrix}, \tag{54}$$

and its corresponding consistent MPR (calculated by Equation (*2*)) is

$$\tilde{K}_1 = \begin{pmatrix} 1 & 0.6117 & 0.2749 & 0.5946 \\ * & 1 & 0.4495 & 0.9721 \\ * & * & 1 & 2.1627 \\ * & * & * & 1 \end{pmatrix}. \tag{55}$$

Similar calculations are carried out for $A_2$ and $A_3$.

From Equation (*10*), one obtains: $CI_{\overset{*}{K_1}}(\overset{*}{A_1}) = 1.66161$, $CI_{\overset{*}{K_2}}(\overset{*}{A_2}) = 1.207088$ and $CI_{\overset{*}{K_3}}(\overset{*}{A_3}) = 1.09$. As a consequence, $A_1$ and $A_2$ are not consistent and $A_3$ is acceptably consistent.

We apply Algorithm IC-I-MPR to $A_1^{(1)} = A_1$ and $A_2^{(2)} = A_2$ with initialization points $x_0^{(1)} = [3/2, 1/5, 1/3, 1/7, 11/2, 1/2]$ and $x_0^{(2)} = [5/2, 1/3, 3/4, 1, 5/2, 3/2]$ which are the midpoints of $A_1$ and $A_2$, respectively.

From Equation (*30*), $\overset{o}{A_1} = A_1$ and $\overset{o}{A_2} = A_2$.

Let us provide the solution for $A_1$ which has the highest CI. Then, from Equation (*33*) and the first expert, the following inequalities will be utilized in the optimization process:

$$\begin{array}{ccccc} 1 - \epsilon_c & \leq & x_1 & \leq & 2 + \epsilon_c, \\ 1/5 - \epsilon_c & \leq & x_2 & \leq & 1/5 + \epsilon_c, \\ 1/3 - \epsilon_c & \leq & x_3 & \leq & 1/2 + \epsilon_c, \\ 1/7 - \epsilon_c & \leq & x_4 & \leq & 1/7 + \epsilon_c, \\ 5 - \epsilon_c & \leq & x_5 & \leq & 6 + \epsilon_c, \\ 1/2 - \epsilon_c & \leq & x_6 & \leq & 1/2 + \epsilon_c, \end{array} \tag{56}$$

where $\epsilon_c = 0.001$, but for the first iteration $\epsilon_c = 0.0$.

*From Equation (37) and variable's definition given by Equation (34), the objective functional is obtained by replacing $a_{12} = x_1$, $a_{13} = x_2$, $a_{14} = x_3$, $a_{23} = x_4$, $a_{24} = x_5$ and $a_{34} = x_6$. Thus, one obtains:*

$$
f(x) = sign * \left[ \frac{1}{16} \left[ \underbrace{\left( \frac{x_2 x_3}{x_1^2 x_4 x_5} \right)^{1/4} + \frac{1}{\beta_1}} + \underbrace{\left( \frac{x_2^2 x_6}{x_1 x_3 x_4} \right)^{1/4} + \frac{1}{\beta_2}} + \underbrace{\left( \frac{x_3^2}{x_1 x_2 x_5 x_6} \right)^{1/4} + \frac{1}{\beta_3}} + \underbrace{\left( \frac{x_4^2 x_1 x_6}{x_2 x_5} \right)^{1/4} + \frac{1}{\beta_4}} + \right. \right.
$$
$$
\left. \left. \underbrace{\left( \frac{x_5^2 x_1}{x_3 x_4 x_6} \right)^{1/4} + \frac{1}{\beta_5}} + \underbrace{\left( \frac{x_6^2 x_2 x_4}{x_3 x_5} \right)^{1/4} + \frac{1}{\beta_6}} + \frac{1}{4} \right] \right];
$$

(57)

*where $\beta_1$ equals the term over the first brace, $\beta_2$ equals the term over the second brace, and so on. In addition, $sign = 1$ is defined to obtain the minimization and $sign = -1$ for the maximization.*

*The same procedure is used for the second expert ($A_2$). After applying the SQP, the results follow:*

$$
\overline{A}_1 = \begin{pmatrix} 1 & 0.5610 & [0.5713\ 0.5890] & 0.9390 \\ * & 1 & 0.5819 & 4.5610 \\ * & * & 1 & 0.9390 \\ * & * & * & 1 \end{pmatrix}, \ \overline{A}_2 = \begin{pmatrix} 1 & 2.5 & 1/3 & 3/4 \\ * & 1 & 1/2 & [2\ 2.0303] \\ * & * & 1 & [1.7628\ 2] \\ * & * & * & 1 \end{pmatrix}.
$$

(58)

*Now that we have the set of I-MPRs in acceptable Individual Consistency, we can proceed with solving the Group Consensus for $\overline{A}_1$, $\overline{A}_2$ and $A_3$.*

*Let us now apply Definition 4 to these I-MPRs. The set $\{ \overset{1}{\check{A}} \}$ is calculated as in Equation (25). For example,*

$$
\overset{1}{\check{A}} = \{ \overset{1}{A}_1, \overset{1}{A}_2, \overset{1}{A}_3 \}; \overset{2}{\check{A}} = \{ \overset{1}{A}_1, \overset{1}{A}_2, \overset{2}{A}_3 \}; \overset{3}{\check{A}} = \{ \overset{1}{A}_1, \overset{2}{A}_2, \overset{1}{A}_3 \}; \cdots; \overset{262143}{\check{A}} = \{ \overset{64}{A}_1, \overset{64}{A}_2, \overset{63}{A}_3 \}, \overset{262144}{\check{A}} = \{ \overset{64}{A}_1, \overset{64}{A}_2, \overset{64}{A}_3 \}.
$$

(59)

*From Definition 5, let us apply it to $\overline{A}_1$, $\overline{A}_2$ and $A_3$ where, by Equation (3), one obtains:*

$$
\overset{1}{\check{A}}{}^c = \left[ \left( a_{ij}^{(1)} \right)^{\lambda_1} \cdot \left( a_{ij}^{(2)} \right)^{\lambda_2} \cdot \left( a_{ij}^{(3)} \right)^{\lambda_3} \right], \ i, j = 1, 2, 3, 4,
$$

(60)

*where $(a_{ij}^{(x)})^{\lambda_x}$, $x = 1, 2, 3$ comes from $\overset{1}{\check{A}}_1$ (cf. Equation (59)), for $\overset{2}{\check{A}}{}^c$ it comes from $\overset{2}{\check{A}}_1$, and so on. Note that, in this example, $\lambda_1 = \lambda_2 = \lambda_3 = 1/3$.*

*From Definition 6, we note that the Group Consensus of each DM gives the following result. Since $GCI_{\check{H}_t} (\check{A}_1) = 1.0985$, $GCI_{\check{H}_t} (\check{A}_2) = 1.0473$ and $GCI_{\check{H}_t} (\check{A}_3) = 1.0381$, all the experts are in acceptable consensus with a global ranking order $w_3 \overset{0.9524}{\succ} w_2 \overset{0.8081}{\succ} w_1 \overset{1}{\succ} w_4$.*

*Since we have detailed the calculations of the Algorithm 1, in the next example, we provide details of the Algorithm 2.*

In the next example, we address and solve a problem of partner selection that is determined for the formation of a virtual enterprise already considered in [51,52].

A virtual enterprise is a dynamic association of enterprises, working together to benefit from a market opportunity by giving a solution that could not be delivered individually. Naturally, it comes with a market opportunity and then it is stopped after that market particular event. Thus, a virtual enterprise is, by definition, a non-permanent alliance of diverse, autonomous, and in some cases geographically distributed organizations sharing resources and skills having common objectives and profiting from a benefit window in the market opportunities [51].

**Example 4.** *In this example, the main enterprise needs to select a partner to grasp a new market opportunity where four candidates (alternatives) must be analyzed by four DMs. The CEO's management staff (DMs) is*

*involved in the partner selection process which should select the most suitable partner. Then, each DM compares each pair of alternatives $c_i$ and $c_j$, and gives her/his preference assessment through the next I-MPRs:*

$$
A_1 = \begin{pmatrix} 1 & [2/3\ 1] & [3/2\ 2] & [5/3\ 2] \\ * & 1 & [1\ 3] & [1\ 2] \\ * & * & 1 & [1/2\ 1] \\ * & * & * & 1 \end{pmatrix}, \quad
A_2 = \begin{pmatrix} 1 & [2\ 3] & [3\ 4] & [2\ 6] \\ * & 1 & [3/2\ 3] & [3\ 4] \\ * & * & 1 & [9/10\ 1] \\ * & * & * & 1 \end{pmatrix},
$$

$$
A_3 = \begin{pmatrix} 1 & [1/2\ 1] & [2\ 3] & [1\ 3] \\ * & 1 & [3\ 4] & [2\ 4] \\ * & * & 1 & [1/4\ 1/2] \\ * & * & * & 1 \end{pmatrix}, \quad
A_4 = \begin{pmatrix} 1 & [1\ 2] & [2/3\ 2] & [2\ 3] \\ * & 1 & [2/3\ 6/5] & [2\ 5] \\ * & * & 1 & [3\ 7/2] \\ * & * & * & 1 \end{pmatrix}, \tag{61}
$$

*where the initial individual order of intervals for each DM is given by*

$DM1: x_2 \overset{1}{\succ} x_1 \overset{1}{\succ} x_4 \overset{1}{\succ} x_3$; $DM2: x_1 \overset{1}{\succ} x_2 \overset{1}{\succ} x_3 \overset{.6698}{\succ} x_4$; $DM3: x_2 \overset{1}{\succ} x_1 \overset{1}{\succ} x_4 \overset{1}{\succ} x_3$ & $DM4: x_3 \overset{.6382}{\succ} x_1 \overset{.9904}{\succ} x_2 \overset{1}{\succ} x_4$.

*In [52], it is found that the third expert is more consistent and then it is assigned a higher weight. They obtained the final ranking of the alternatives as: $x_2 \succ x_1 \succ x_4 \succ x_3$. On the other hand, when all experts are considered equally weighted, they obtained the final ranking of the alternatives as $x_1 \succ x_2 \succ x_3 \succ x_4$.*

*Let us apply our method to check for consistency and consensus indices.*

*By applying the first part of our Definitions, the Individual Consistency of each DM (by using Equation (12)) is given by:*

$$
CI_{A_1} = 1.060286, \quad CI_{A_2} = 1.076585, \quad CI_{A_3} = 1.101763, \quad CI_{A_4} = 1.078917, \tag{62}
$$

*where it is noted that the third DM ($A_3$) has provided a slightly inconsistent I-MPR.*

*Since the third expert's judgments $A_3$ are not consistent, let us apply the **Algorithm IC-I-MPR**.*

*Take an initialization point $x_0^{(1)} = [3/4, 5/2, 2, 7/2, 3, 3/4]$ and $\epsilon_c = 0.001$ and the result follows:*

$$
\overline{A}_3 = \begin{pmatrix} 1 & [0.7413\ 1] & [2\ 2.9656] & [1.4825\ 2.9962] \\ * & 1 & [3\ 4] & [2\ 3.8970] \\ * & * & 1 & [1/4\ 1/2] \\ * & * & * & 1 \end{pmatrix}. \tag{63}
$$

*Now that we have the whole set of I-MPRs ($A_1$, $A_2$, $\overline{A}_3$ and $A_4$) in acceptable consistency, we can proceed to address the group consensus analysis.*

*First case: Although the first expert has the most consistent I-MPR (cf. Equation (62)), let us consider that the third expert $A_3$ is the more relevant, by assigning the next following experts' weighting $\lambda = [1/5, 1/5, 2/5, 1/5]$.*

*From Definition 6 applied to the first DM ($A_1$), it follows:*

$$
GCI_{\check{H}_1}(\check{A}_1) = \max\{ GCI_{\underset{\check{A}^c}{1}}(\check{A}), GCI_{\underset{\check{A}^c}{2}}(\check{A}), \cdots, GCI_{\underset{\check{A}^c}{\nu-1}}(\overset{\nu-1}{\check{A}}), GCI_{\underset{\check{A}^c}{\nu}}(\overset{\nu}{\check{A}}) \}, \tag{64}
$$

*where in this case $\nu = 16777216$. Similar calculations are carried out for $A_2$, $\overline{A}_3$ and $A_4$.*

*Then, we obtain $GCI_{\check{H}_1}(\check{A}_1) = 1.0657$, $GCI_{\check{H}_2}(\check{A}_2) = 1.0646$, $GCI_{\check{H}_3}(\check{A}_3) = 1.0908$, and finally $GCI_{\check{H}_4}(\check{A}_4) = 1.5364$. Thus, the fourth expert is not in acceptable consensus.*

From Equation (*31*) $A_1^p = A_4$ and from Equation (*33*) the following inequalities will be utilized in the optimization process:

$$
\begin{array}{ccccc}
1 - \epsilon_g & \leq & x_1 & \leq & 2 + \epsilon_g, \\
2/3 - \epsilon_g & \leq & x_2 & \leq & 2 + \epsilon_g, \\
2 - \epsilon_g & \leq & x_3 & \leq & 3 + \epsilon_g, \\
2/3 - \epsilon_g & \leq & x_4 & \leq & 6/5 + \epsilon_g, \\
2 - \epsilon_g & \leq & x_5 & \leq & 5 + \epsilon_g, \\
3 - \epsilon_g & \leq & x_6 & \leq & 7/2 + \epsilon_g,
\end{array}
\tag{65}
$$

where $\epsilon_g = 0.001$, but for the first iteration $\epsilon_g = 0.0$.

From Equation (*39*) and variable's definition given by Equation (*34*), the objective functional is obtained by replacing in it $a_{12}^{(4)}$, $a_{13}^{(4)}$, $a_{14}^{(4)}$, $a_{23}^{(4)}$, $a_{24}^{(4)}$ and $a_{34}^{(4)}$. Then, one obtains:

$$
\begin{aligned}
f(x) = \ sign * \Bigg[ \frac{1}{16} \Bigg[ & \left( \frac{x_{19}^{(1-\lambda_4)}}{a_{12}^{(1)\lambda_1} a_{12}^{(2)\lambda_2} a_{12}^{(3)\lambda_3}} \right) + \frac{1}{\alpha_1} + \left( \frac{x_{20}^{(1-\lambda_4)}}{a_{13}^{(1)\lambda_1} a_{13}^{(2)\lambda_2} a_{13}^{(3)\lambda_3}} \right) + \frac{1}{\alpha_2} + \left( \frac{x_{21}^{(1-\lambda_4)}}{a_{14}^{(1)\lambda_1} a_{14}^{(2)\lambda_2} a_{14}^{(3)\lambda_3}} \right) + \frac{1}{\alpha_3} + \\
& \left( \frac{x_{22}^{(1-\lambda_4)}}{a_{23}^{(1)\lambda_1} a_{23}^{(2)\lambda_2} a_{23}^{(3)\lambda_3}} \right) + \frac{1}{\alpha_4} + \left( \frac{x_{23}^{(1-\lambda_4)}}{a_{24}^{(1)\lambda_1} a_{24}^{(2)\lambda_2} a_{24}^{(3)\lambda_3}} \right) + \frac{1}{\alpha_5} + \left( \frac{x_{24}^{(1-\lambda_4)}}{a_{34}^{(1)\lambda_1} a_{34}^{(2)\lambda_2} a_{34}^{(3)\lambda_3}} \right) + \frac{1}{\alpha_6} \Bigg] + \frac{1}{4} \Bigg],
\end{aligned}
\tag{66}
$$

where $\alpha_1$ equals the term over the first brace, $\alpha_2$ equals the term over the second brace, and so on. In addition, $sign = 1$ is defined to obtain the minimization and $sign = -1$ for the maximization.

Let us then apply the SQP algorithm for $A_4^4 = \left( \overset{-(4)}{a_{ij}} \ \overset{+(4)}{a_{ij}} \right)$ with $\overline{GCI} = 1.1$ and $\epsilon = 0.001$.

As soon as the optimization algorithm converges, the new Group Consensus Indices from Equation (*23*) is read as:

$$
GCI_{\breve{A}_1} = 1.0132, \quad GCI_{\breve{A}_2} = 1.0921, \quad GCI_{\breve{A}_3} = 1.0508, \quad GCI_{\breve{A}_4} = 1.0697.
\tag{67}
$$

In addition, from Equation (*24*):

$$
GCI_{\overline{A}_1} = 1.0008, \quad GCI_{\overline{A}_2} = 1.0722, \quad GCI_{\overline{A}_3} = 1.0495, \quad GCI_{\overline{A}_4} = 1.0777.
\tag{68}
$$

The priority vector is $w_1 = [0.268889 \ 0.459874]$, $w_2 = [0.284454 \ 0.400261]$, $w_3 = [0.09285 \ 0.186909]$, $w_4 = [0.123878 \ 0.2380]$, and the final ranking of priorities is given by $w_1 \overset{.6153}{\succ} w_2 \overset{1}{\succ} w_4 \overset{0.8149}{\succ} w_3$, or $x_1 > x_2 > x_4 > x_3$.

Second case: For an equal experts' weighting (For example, $\lambda = [1/4, 1/4, 1/4, 1/4]$), one has that the Group Consensus Indices for each decision maker (by using Equation (*23*)) are:

$$
GCI_{\breve{A}_1} = 1.0902, \quad GCI_{\breve{A}_2} = 1.0597, \quad GCI_{\breve{A}_3} = 1.1458, \quad GCI_{\breve{A}_4} = 1.4162,
\tag{69}
$$

where the third and fourth DM have provided their I-MPRs not in consensus.

Then, apply again **Algorithm GC-I-MPR** for $A_3^3$, $A_4^4$, $\overline{GCI} = 1.1$ and $\epsilon = 0.001$, but this time with the new $\lambda = [1/4, 1/4, 1/4, 1/4]$.

As soon as the optimization algorithm converges, the new Group Consensus Indices from Equation (*23*) is read as:

$$
GCI_{\breve{A}_1} = 1.0436, \quad GCI_{\breve{A}_2} = 1.0528, \quad GCI_{\breve{A}_3} = 1.0983, \quad GCI_{\breve{A}_4} = 1.0463.
\tag{70}
$$

In addition, from Equation (*24*):

$$
GCI_{\overline{A}_1} = 1.0058, \quad GCI_{\overline{A}_2} = 1.0348, \quad GCI_{\overline{A}_3} = 1.0365, \quad GCI_{\overline{A}_4} = 1.0590,
\tag{71}
$$

where the set of I-MPRs are Individually Consistent and in an acceptable Group Consensus. They are read as:

$$
A_1 = \begin{pmatrix} 1 & [0.7507\ 1.0] & [1.6237\ 2.0] & [1.8408\ 2.0] \\ * & 1 & [1.1212\ 3.0] & [1.1860\ 2.0] \\ * & * & 1 & [0.5505\ 1.0] \\ * & * & * & 1 \end{pmatrix},
$$

$$
A_2 = \begin{pmatrix} 1 & [2.0\ 2.1349] & [3\ 3.1074] & [2.0\ 4.3848] \\ * & 1 & [3/2\ 2.5193] & [3.0\ 3.1426] \\ * & * & 1 & [9/10\ 0.9112] \\ * & * & * & 1 \end{pmatrix},
$$

$$
A_3 = \begin{pmatrix} 1 & [0.9092\ 1.0] & [2.0\ 2.5686] & [1.7908\ 2.9962] \\ * & 1 & [2.9035\ 3.0] & [2.0\ 3.1838] \\ * & * & 1 & [0.388\ 1/2] \\ * & * & * & 1 \end{pmatrix},
$$

$$
A_4 = \begin{pmatrix} 1 & [1.0\ 1.4055] & [1.4248\ 2.0] & [2.2640\ 2.4575] \\ * & 1 & [1.3641\ 1.3654] & [2.0\ 2.9146] \\ * & * & 1 & [1.7183\ 1.8753] \\ * & * & * & 1 \end{pmatrix}.
$$

(72)

*Their priority vector is $w_1 = [0.308967\ 0.488215]$ $w_2 = [0.25554\ 0.397948]$, $w_3 = [0.104711\ 0.235793]$, $w_4 = [0.113979\ 0.198821]$, and the final ranking of priorities is given by $w_1 \overset{0.8449}{\succ} w_2 \overset{1}{\succ} w_3 \overset{0.6057}{\succ} w_4$, or $x_1 > x_2 > x_3 > x_4$.*

*We note that, through Definitions, Theorem and Algorithms introduced here, we obtain the same results of [51,52] when the experts' weight is the same. However, when the weighting of the experts is different, some small differences are found. The above is due to the variation in the weighting provided here and in those articles.*

*Case Study Discussions and Managerial Implications*

In the first example, the second and third DMs have provided inconsistent I-MPRs, and it was sufficient to improve their Individual Consistency to have the three I-MPRs in Group Consensus. Note that the ranking order of each DM was initially slightly different among them; nevertheless, at the end of the process, the Group Consensus points to an acceptable and analytic decision.

As we have have seen along the first example, when a group of DMs needs to state a point of agreement, they can define their assessments through I-MPRs which do not necessarily need to be exactly in the same judgment direction. Sometimes, they only need to be confident in the quality of their decisions based on an MCDM system which analyzes them through a well-known decision model.

For the second example, only two DMs evaluate in the same direction the set of criteria (alternatives). The other two experts are even in clear contradiction, since the first DM states that the third criteria is lesser in importance and the second DM states that this criteria is the most important. Furthermore, by assigning to the third expert a highest evaluation confidence (weight), the methodology produced a very interesting result, by selecting a global ranking order that none of them had chosen.

On the other hand, when every DM has the same weight, the methodology produced also a different global ranking order that none of them had chosen. The DMs $A_1$, $A_2$ and $A_3$, have preserved their individual ranking order at the end of the process, and only the fourth DM has had his individual ranking order changed. This result can be used in his/her operating department as an internal feedback to reconsider their position with respect to the other departments and the organization objectives.

As soon as a DM obtains the results, s/he can use them so that, with this information, he can state new ways of organizing her/his operating department. For example, this method can also be carried out within the operating departments since each one of their business operations have criteria and alternatives that can be better emphasized.

In summary, the MCDM methodology can provide results which could reinforce the position of a set of DMs or point out to a new direction.

## 6. Concluding Remarks and Future Work

In this paper, we provide a methodology based on a couple of algorithms and a nonlinear optimization approach to be used when a heterogeneous managers group needs to solve an MCDM problem.

Our approach can use Interval Multiplicative Preference Relations or Multiplicative Preference Relations, and demonstrates the utilization of the methodology to synthesize reliable intervals where consistency and consensus constraints hold. Once decision makers have proposed their I-MPRs, our method can solve for these I-MPRs from well-known decision support models. One advantage of our algorithm is that DM can re-express their preferences within an interval where, usually, they have to observe some constraints based on decision targets, framework rules and advice. When the DM is confident on the pair of criteria (o alternatives) under evaluation, s/he can utilize a crisp value. On other other hand, when s/he is hesitant or uncertain about the assessment, s/he could use an interval. Our algorithm can solve independently of the used approach.

In this work, reliable I-MPRs provide a distinct advantage in interpretation of hesitancy and uncertainty about the final consistency and consensus.

Main advantages of the present approach:

- It is provided through a couple of algorithms and a nonlinear optimization approach (Sequential Quadratic Programming) concurrently applied.
- Through the Hadamard's operator and some easy algebraic manipulations, objective functionals were synthesized to be used in the optimization algorithm.
- When the I-MPRs improved by the methodology are reduced into an MPR (defined in the I-MPR), our approach can still give reliable results. For example, for this MPR, we can verify the results of IC or GC with an alternative method.
- The IC or the GC accepted indices (threshold values) have been previously investigated and fixed. Nevertheless, the project designer could assign a different value depending on the project requirements.
- Obtained results are independent of the method of prioritization utilized in the consensus operation.

Main drawbacks of the present approach:

- The computational cost increases as the I-MPRs dimension and the number of DMs involved in the evaluation process are increased.
- For a real project where a high number of criteria and experts participate, it can be necessary to program this method through an exhaustive parallel computation system.
- For a real project where a high number of criteria and experts participate, the notation can be cumbersome.

Future works aim to make an implementation on:

- The application of our approach to various study cases where heterogenous groups of DMs with different weights participate in a collaborative manner.
- The integration of the complete methodology in a benchmark to compare the results of a diverse set of MCDM tools.
- The definition or employment of this methodology on different frameworks, v.gr. fuzzy or hesitant MCDM.

**Funding:** This research was funded by PFCE-PRODEP grant number DIP/DI/CI/2015-1160.

**Acknowledgments:** I would like to thank the anonymous reviewers and the Journal's staff for their constructive advice and support for improving this paper.

**Conflicts of Interest:** The author declares no conflict of interest.

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
