# Peer review of "Multiple Criteria Decision-Making in Heterogeneous Groups of Management Experts"

_information, doi:10.3390/info9120300_

Reviewer 1 Report

Thank you for inviting me as a reviewer for manuscript titled MCDM in Heterogeneous Groups of Management Experts. This paper presents interesting application  of multi criteria decision-making in heterogeneous group decision making. I must congratulation to the authors for effort they made and provided results (the results are good and clear). In my opinion the paper is almost ready for the publication. The paper would be more exiting if you implement below improvements:

- Author should not use abbreviation in tile - Multi Criteria Decision-Making (MCDM).

- Line 1-20 - Abstract needs to be shorten. Abstract is well written and includes introduction, problem statement, methodology, contributions and results, but abstract should be presented in no more then 250-300 words.

- In introduction section authors should clearly present gap in considered scientific area. Please clearly summarise what specific advantages brings your approach. Try to specific that and build a case for your research. Please, focus on novelty. This should be presented in at least one paragraph.

- What are the advantages associated with the proposed approach – This should be clarified to the readers.

- In introduction section author should present a brief literature review of application of MCDM methods in group decision making. Please, provide relevant references in last two years, authors should refer to: DEMATEL-AHP multi-criteria decision making model for the selection and evaluation of criteria for selecting an aircraft for the protection of air traffic. Decision Making: Applications in Management and Engineering, 1 (2), 93-110. https://doi.org/10.31181/dmame1802091p; Evaluation of the railway management model by using a new integrated model DELPHI-SWARA-MABAC. Decision Making: Applications in Management and Engineering, 1 (2), pp. 34-50. https://doi.org/10.31181/dmame1802034v; A multicriteria model for the selection of the transport service provider: A single valued neutrosophic DEMATEL multicriteria model. Decision Making: Applications in Management and Engineering, 1 (2), 121-130. https://doi.org/10.31181/dmame1802128l.

- Section 5 (page 18 and 22) – authors presented impressive results, but authors should provide detail calculations for proposed example. Try in a better way to show the results. Kindly explain your proposed approach steps with the numerical example steps in details. All calculations should be presented. That will help to the readers to better understanding and future implementation this very innovative approach.

- In section 5 authors should provide more comparisons with existing approaches. The case study and the discussion of the results are interesting and appreciated. Provide detail discussion what your model brings more then existing models in MCDM literature (AHP model). What did you bring more? Add more detailed discussion in this section.

- Add Managerial implications section of proposed approach.

- Please summarise the advantages and limitations of the proposed method in practical applications.

- The authors are urged to give more proof and explanation about the validity or practicability of the proposed method.

I will review the final version of the paper with pleasure. Congrats to the Author.

Author Response

Dear Professor, I really appreciate your comments and advice and respectfully I answer your concerns in the following:

In introduction section, the specific advantages has      been added, which one can obtain through the presented approach. Besides,      in the numerical examples the study case has been enlightened and a      discussion section has been provided.

In introduction section has been added the specific      advantages which one can obtain through the presented approach.

A literature review of application of MCDM      methods in group decision making is now enlarged and your contributions have      been taken into consideration in the main literature.

The examples’ calculations has now been improved      and detailed. In consequence the algorithms has been slightly modified to      follows these calculations.

A novel section addressing discussions about the advantages and the      differences of the proposed approach and the literature publications, are      now enlightened. The case study discussions are now integrated to the      paper and a deeper analysis is included.

For the sake of convenience of reading instead of adding a new complete      section, a subsection at the end of Section 5 is now included, where      managerial implications are provided.

The paper includes now the advantages and limitations of the proposed method in practical      applications.

In the Subsection 5.1 and in Section 6 the validity and      practicability of the proposed method is detailed and the main drawbacks      of the proposed approach is provided and analyzed when we want to use this      approach in a real scenario.

We really  appreciate your advice and comments

Reviewer 2 Report

The paper is really interesting and well written scientific work. In my opinion, there is only one serious shortcoming. The literature reviews is insufficient. I suggest extend the part about recent work in the MCDM methods area, e.g.,

--Liu, J., et al. (2017). Decision process in MCDM with large number of criteria and heterogeneous risk preferences. Operations Research Perspectives, 4, 106-112.

-- Zhang, X., et al. (2014, March). The Extended TOPSIS Method for Multi-criteria Decision Making Based on Hesitant Heterogeneous Information. In 2014 2nd International Conference on Software Engineering, Knowledge Engineering and Information Engineering (SEKEIE 2014)). Atlantis Press.

-- Faizi, S.; et al. (2017) Group Decision-Making for Hesitant Fuzzy Sets Based on Characteristic Objects Method. Symmetry, 9, 136

-- Cheng, J., et al. (2018). Structural Optimization of a High-Speed Press Considering Multi-Source Uncertainties Based on a New Heterogeneous TOPSIS. Applied Sciences, 8(1), 126.

-- Faizi, S., et al. (2018). Decision making with uncertainty using hesitant fuzzy sets. International Journal of Fuzzy Systems, 20(1), 93-103.

-- Wątróbski, J., et al. (2018). Generalised framework for multi-criteria method selection. Omega.

-- etc.

Please extend the list of futree works (more descriptive)

Author Response

Dear Professor, I really appreciate your comments and advice and respectfully I answer your concerns in the following:

1.     A literature review of application and approaches of MCDM methods in group decision making is now enlarged and your contributions have been taken into consideration in the main literature. Thanks for your advice.

2.     The list of future works is now included with more detail. Thanks for your advice.

We really appreciate your advice and comments.

Round  2

Reviewer 1 Report

I am very happy that the authors have addressed my concerns point by point precisely. No further suggestions come from my side. Therefore, I would like to recommend this manuscript to be published.